# Proteome Profile Changes Induced by Heterologous Overexpression of *Mycobacterium tuberculosis*-Derived Antigens PstS-1 (Rv0934) and Ag85B (Rv1886c) in *Mycobacterium microti*

**DOI:** 10.3390/biom12121836

**Published:** 2022-12-08

**Authors:** Viridiana García-Ruiz, Patricia Orduña, Antonia I. Castillo-Rodal, Teresa J. Flores-Rodríguez, Yolanda López-Vidal

**Affiliations:** 1Programa de Inmunología Molecular Microbiana, Departamento de Microbiología y Parasitología, Facultad de Medicina, Universidad Nacional Autónoma de México, Av. Ciudad Universitaria 3000, Coyoacán, Ciudad de México 04510, CP, Mexico; 2Laboratorio de Microbioma, División de Investigación, Facultad de Medicina, Universidad Nacional Autónoma de México, Av. Ciudad Universitaria 3000, Coyoacán, Ciudad de México 04510, CP, Mexico

**Keywords:** vaccine, antigen, glycosylation, *Mycobacterium microti*, *Mycobacterium tuberculosis*, BCG, proteome profile

## Abstract

The development of new tuberculosis vaccines remains a global priority, and recombinant vaccines are a frequently investigated option. These vaccines follow a molecular strategy that may enhance protective efficacy. However, their functional differences, particularly with respect to glycosylation, remain unknown. Recent studies have shown that glycosylation plays a key role in the host-pathogen interactions during immune recognition. The aim of this study was to determine the differences in the glycosylation profiles of two recombinant strains of *Mycobacterium microti,* overexpressing Ag85B (Rv1886c) and PstS-1 (Rv0934) antigens of *M. tuberculosis*. For each strain, the glycosylation profile was determined by Western blotting with lectins. The results showed the presence of mannosylated proteins and evidence of linked sialic acid proteins. Interestingly, different proteome and glycoproteome profiles were observed between the two recombinant strains and the wild-type strain. We have shown here that the construction of the recombinant strains of *M. microti* has altered the proteome and glycosylation profiles of these strains, leading us to ask what impact these changes might have on the immune response.

## 1. Introduction

Tuberculosis (TB) has been officially recognized as a global emergency since 1993 [1,2] and is caused by airborne intracellular bacilli of the *Mycobacterium tuberculosis* (Mtb) complex [3]. According to the Global Tuberculosis Report 2019, TB remains one of the top ten causes of death worldwide [4] and is the leading cause of death associated with an infectious disease in adults [1], particularly in low-income countries [2]. Every day, nearly 4000 people die due to TB [4,5], and part of this high number is due the lack of effective vaccines and diagnostic tools [6,7].

The bacillus Calmette-Guerin (BCG) of *Mycobacterium bovis* is currently the only vaccine available worldwide for the prevention of tuberculosis, with a protective efficacy of up to 80% against severe forms of TB (i.e., miliary and meningeal) [8,9]. However, against pulmonary TB, the most common form of the disease, this vaccine is only 50–80% effective [10]. These variations in the protective efficacy have inspired efforts to develop new vaccines, including subunit vaccines [11,12,13], live vaccines against bacteria within the same complex [14] and recombinant vaccines consisting of immunogenic antigens [15,16]. So far, none of these vaccines has successfully elicited a better immune response than the BCG vaccine [17,18].

Other attempts to achieve better results than the BCG vaccine have been investigated, in particular live bacterial vaccines, with *Mycobacterium vaccae* or *Mycobacterium microti,* which belong to the *Mycobacterium tuberculosis* complex (MTBC) [19,20]. *M. microti* causes pulmonary TB in voles but is naturally attenuated in humans [19]. Between 1950 and 1969, this strain was used as a TB vaccine, and showed approximately a 77% protective efficacy with the same immunogenicity against TB [21]. However, about 3–17% of patients that were vaccinated with *M. microti* developed skin reactions with a more lupoid aspect and a greater severity than those vaccinated with BCG, which led to discontinuation of the *M. microti* vaccine [22]. This effect was attributed to the administration of the *M. microti* vaccine with multiple pricks. *M. microti* has been used as a vaccine in animal models and for alternative routes of administration, such as intraperitoneal, orogastric and intravenous administration. These studies demonstrated that *M. microti* conferred similar protection levels as the post-infection BCG vaccine when tested against the *M. tuberculosis* strain CDC1551 in mice [19,23].

The increase in multi-drug resistance of Mtb compared to classical treatments such as rifampicin and isoniazid [24,25], estimated to cost USD 13 billion annually worldwide [4], has made the development of more effective vaccines and diagnostics more urgent. In this sense, the research into Mtb virulence factors such as immunogenic antigens are attractive, as they have shown promising results when used for these purposes [18].

Mtb produces several antigenic proteins, some of which are post-translationally modified, notably by mannose type glycan attachments [26,27,28,29] which contribute to disease persistence by allowing bacteria to colonize and invade the host cell [28,30,31,32]. These secreted Mtb proteins, including PstS-1 and Ag85B, trigger antigen-specific host immune responses [33]. Korycka-M et al., postulated that the PPE51 protein participates in the uptake of disaccharides from *M. tuberculosis* [34].

PstS-1 (Rv0934) is a 38 kDa cell wall glycol lipoprotein with attached mannoses. This antigen has been described as an adhesin that induces the proliferation of CD4+ T cells and induces the expression of high levels of interferon (IFN)-γ and immunoglobulin (Ig)G2a [35]. In addition, PstS-1 induces the upregulated production of interleukin (IL)-12 by macrophages and stimulates the CD8+ cytotoxic T-cells [36,37,38,39].

Ag85B (Rv1886c) is a 30 kDa mycolil transferase that plays a role in the synthesis of the cell wall components. It is one of the main components of the culture filtrates in both the *M. tuberculosis* and *M. bovis* BCG vaccine [40]. Ag85B also belongs to a group of three proteins known as complex 85, which is a component of the cell wall in all species of mycobacteria [41]. This antigen has been extensively studied and tested during the production of DNA, protein and recombinant subunit vaccines [42,43,44,45].

Glycosylation is a post-translational modification (PTM) and refers to the covalent linkage between carbohydrates and a polypeptide, lipid, polynucleotide or organic compound [46,47,48,49]. In Mtb, this PTM is involved in various cellular mechanisms, including pathogenesis, virulence and antigenic properties such as the stimulation of the immune response against various pathogens [31,39,50].

The use of mycobacterial glycoproteins is an option for the development of a TB vaccine. Studies have shown that glycosylation plays a key role in the T-cell antigenicity, suggesting that the use of glycosylated proteins in vaccines could enhance the induction of a protective immune response [51]. This effect has been associated, for example, with the native glycosylation pattern of APA (Rv1860), also known as 45/47 kDa. When the pattern is altered, a lower stimulation of T-lymphocyte responses is observed [52,53]. However, there is little research about the role of glycosylation on recombinant antigens and its importance in the immunological response against TB.

The first glycosylated proteins in *M. tuberculosis* were described in the 1970s and 1980s, and recent quantitative glycoproteomic analyses have revealed 101 differentially glycosylated sites in 67 proteins in four *M. tuberculosis* strains [54]. Most proteins that have been described for Mtb are O-mannosylated proteins. However, the presence of other types of glycosylation, glycan types, glycan occupancy and glycan length are poorly understood.

Most of the identified glycosylated proteins/lipids in mycobacteria are located in the cell wall or are secreted. These glycoproteins are involved in several biological mechanisms, such as cell adhesion, invasion, cellular signaling, intracellular bacterial survival, antigenicity, pathogenicity and virulence, among others [50,54]. Nandakumar et al. (2013) have shown that mannosylation of the adhesin APA in *M. tuberculosis* is essential for the human T-cell activation in a murine model of subcutaneous infection. In addition, the differences in cytokine production, including IFN-γ, the tumor necrosis factor (TNF)-α and IL-2, have been demonstrated following the immunization with the APA protein, independent of the mannosylated state [27]. The importance of glycosylation for various *M. tuberculosis* proteins and the effects of glycosylation on the interactions with the immune response need to be further explored to understand their implications for vaccine development [51].

Recently, our research group studied the *M. microti* strain ATCC 11,152 in a murine model of pulmonary TB, and we observed that the bacterial burden of *M. tuberculosis* was significantly reduced, resulting in a lower percentage of pneumonia in lung tissue after it was challenged with *M. tuberculosis* H37Rv (data not shown). Based on these results and the positive outcomes of recombinant vaccines, we decided to use *M. microti* as a vehicle for the overexpression of the *M. tuberculosis* antigens PstS-1 and Ag85B. These antigens have been widely studied and are known to stimulate T- and B-lymphocyte proliferation and elicit a T-helper (Th)1-type response in animal models [37,38,55,56,57]. These antigens are encoded in the genome of *M. microti*, but little is known about the in vitro and in vivo expression of the two antigens in this strain.

In the present study, we developed two recombinant strains of *M. microti*, overexpressing these PstS-1 and Ag85B antigens. The protein expression of these antigens was performed in a constitutive vector and analyzed with SDS-PAGE and 2D-PAGE. In addition, the proteomic profiles of the developed strains and the wild-type strain were examined and the differences in the proteomic profiles between the three strains were found, suggesting that the recombination event not only affected the protein expression but also changed the global proteomic profile of these strains.

## 2. Materials and Methods

### 2.1. Reagents

The culture media were purchased from Thermo Fisher Scientific™ (BD Difco™ Middlebrook 7H10 Agar and Middlebrook 7H9 Broth Base, Yeast Extract, Bacto™ Triptone; Becton Dickinson, Detroit, MI, USA), and JT Baker, Phillipsburg, NJ, USA (NaCl, KCl, MgSO_4_, CaCl_2_, Glycerol, KOAc, Glucose, l-Asparagine, KH_2_PO_4_, Ferric Ammonium Citrate, Citric Acid, Sodium Pyruvate).

The molecular biology reagents were purchased from Promega™, Madison, WI, USA (Wizard^®^ SV Gel and PCR Clean-Up System), SigmaTM, St Louis, MO, USA (Ethidium Bromide, Agarose, Kanamycin, Tween^®^ 80), InvitrogenTM, Carlsbad, CA, USA (TOPO™ Cloning Kit with pCR™2.1, Anza™ T4 DNA Ligase Master Mix) and New England BioLabs^®^, Ipswich, MA, USA (BamHI, EcoRI, HindIII). The pMV261 plasmid was provided by Clara Espitia and Luis Servín (Instituto de Investigaciones Biomédicas, UNAM, Mexico).

The proteomic reagents were purchased from Bio-Rad, Hercules, CA, USA (Bradford Assay, Bio-Lyte^®^ 3/10 Ampholyte, ReadyStrip™ IPG Strips) and Amersham Biosciences, Buckinghamshire, UK (CyDye™ DIGE Fluor Labelling Kit).

The following reagents were obtained through TBVTRM Contract, Colorado State University, USA: purified protein PstS1 from *Mycobacterium tuberculosis* (NR-14859), monoclonal anti-*Mycobacterium tuberculosis* PhoS1/PstS1 (NR-13605) and purified protein Ag85B from *Mycobacterium tuberculosis* H37Rv (NR-53526).

### 2.2. Strains

For the transformation and expression of the antigens PstS-1 (Rv0934) and Ag85B (Rv1886c), we used *Mycobacterium microti* ATCC 11152, provided by Dr. Gutiérrez-Pabello (Faculty of Medicine UNAM). For the DNA extraction and cloning, we used the strain *Mycobacterium tuberculosis* H37Rv provided by Dra. Torres (INER). For the propagation, cloning and subcloning of the constructed vectors, we used *E. coli* DH5α competent cells provided by Dr. L. Servín (Instituto de Investigaciones Biomédicas, UNAM, Mexico).

### 2.3. DNA Extraction and Sequencing

The PstS-1 (Rv0934) and Ag85B (Rv1886c) coding genes in *M. microti* and *M. tuberculosis* were amplified by a polymerase chain reaction (PCR) and submitted to sequencing. Initially, *M. microti* and *M. tuberculosis* were grown in BD Difco™ Middlebrook 7H10 agar and incubated at 37 °C for 15 days. After the appearance of colonies, a single colony of each strain was taken and inoculated into 100 mL of Middlebrook 7H9 broth base supplemented with 0.05% Tween 80 and 10% ADC (Albumin-dextrose-catalase) in flasks, allowed to grow at 37 °C and shaken at 150 rpm until an A.U. of 0.6 (O.D. 600 nm) was achieved. The cells were harvested using centrifugation at 10,000 rpm for 15 min at 4 °C. The supernatant was discarded, and the pellet resuspended in 10 mL of glycerol 10%. The genomic DNA was extracted using the lysozyme and proteinase K method [58]. The DNA was resuspended in 50 µL of distilled H_2_O and stored at −20 °C for further use. Specific primers were designed for each gene based on the genome of *M. tuberculosis* H37Rv (GeneBank: AL123456.3). For PstS-1 (Rv0934), the forward primer 5′-ATAGTACGAGGATCCTGAAAATTCGTTTGC-3′ and the reverse primer 5′-ATAGTACGAGAATTCCTAGCTGGAAATCGT-3′ were used with the restriction sites BamHI and EcoRI (underlined). For Ag85B (Rv1886c), the forward primer 5′-GGCTGAATTCATGACAGACGTGAGCCGAAA-3′ and the reverse primer 5′-CGCGAAGCTTCAGCCGGCGCCTAACGAAC-3′ were used with the restriction sites EcoRI and HindIII (underlined). The PCR conditions for PstS-1 (Rv0934) were used as follows: one initial denaturation cycle at 94 °C followed by 30 cycles of 1 min at 94 °C for denaturation, 1 min at 56 °C for annealing and 1 min at 72 °C for elongation. For Ag85B (Rv1886c), there was an initial denaturation cycle at 94 °C and then 35 cycles of 1 min at 94 °C for denaturation, 1 min at 55 °C for annealing and 1 min at 72 °C for elongation. The PCR amplicons were visualized with a 2% gel electrophoresis and stained with ethidium bromide. They were then cleaned using the Wizard^®^ SV Gel and the PCR Clean-Up System and subcloned using a TOPO™ cloning kit with pCR™2.1 according to the manufacturer’s instructions. Subsequently, the resulting vector constructs were sequenced using commercially available primers: M13 in an ABI PRISM 3100 Genetic Analyzer (Applied Biosystems™, Foster City, CA, USA) at the Biology Institute, UNAM. The sequences were assembled using the SnapGene^®^ viewer 6.1 bioinformatics software (Dotmatics, Boston, MA, USA) and analyzed using the Basic Local Alignment Search Tool (BLAST; NCBI, Bethesda, MD, USA) and the Clustal Omega [59] bioinformatics tools. The constructed vectors of *M. microti*-TOPO-PstS-1, *M. microti*-TOPO-Ag85B, *M. tuberculosis*-TOPO-PstS1 and *M. tuberculosis*-TOPO-Ag85B were transformed into the competent *E. coli* DH5α by heat shock using standard procedures [60] and stored at −20 °C for further use.

### 2.4. Vector Construction

The vectors *M. tuberculosis*-TOPO-PstS1 and *M. tuberculosis*-TOPO-Ag85B were purified from *E. coli* GM2199 and digested with the restriction enzymes BamHI/EcoRI and EcoRI/HindIII, respectively, to obtain the desired sequences. The PstS-1 and Ag85B gene inserts were then resolved with 1% agarose electrophoresis and purified using the Wizard^®^ SV Gel and the PCR Clean-Up System. A reaction of the previously BamHI/EcoRI and EcoRI/HindIII digested vectors pMV261 was ligated with the purified gene inserts PstS-1 and Ag85B using the Anza™ T4 DNA Ligase Master Mix according to the manufacturer’s instructions. The correct cloning of the inserts in the vectors designated pMV261-PstS-1 and pMV261-Ag85B, in which the promoter was Hsp60, was confirmed by the sequencing and then transformed into the competent *E. coli* DH5α as described above. The isolation of the plasmids from the *E. coli* strains was performed using the alkaline lysis method. The transformation of the plasmids pMV261 (control), pMV261-PstS-1 and pMV261-Ag85B into *M. microti* was achieved by electroporation. Briefly, 1 µg of purified DNA from each plasmid was added to the competent *M. microti* cells, and the mixture was incubated on ice for 10 min. The mixture was then placed in a Gene Pulser™ 0.2 cm electroporation cuvette (Bio-Rad, Richmond, CA, USA) and a 2.5 kV current pulse was applied. The cuvette was placed on ice for 4 min. Then 1 mL of Middlebrook 7H9 broth base was added for both strains. The culture was incubated overnight at 37 °C with stirring at 150 rpm. Finally, the cells were harvested using centrifugation and the different dilutions were spread on the BD Difco™ Middlebrook 7H10 Agar containing 30 µg/mL kanamycin. The plates were incubated at 37 °C for 3 to 4 weeks until the presence of colonies was observed. The colonies were selected and the corresponding inserts were confirmed by sequencing, as described above. The strains were designated *M. microti*-pMV261 (as control), *M. microti*-PstS-1 and *M. microti*-Ag85 and cryopreserved at −80 °C for further experiments.

### 2.5. Antigen Expression in M. microti Strains

The strains *M. microti*-pMV261, *M. microti*-PstS-1 and *M. microti*-Ag85 were grown in 250 mL flasks containing 50 mL of Sauton media (composition per liter: 4 g l-asparagine, 0.5 g monopotassium phosphate, 0.5 g magnesium sulphate, 50 mg ferric ammonium citrate, 2 g citric acid, 60 mL glycerol and 0.05% Tween 80), supplemented with 40 mM of glucose and 40 mM of sodium pyruvate at 37 °C with constant stirring at 150 rpm. The kinetic growth was determined every 48 h for 196 h by measuring the O.D. (600 nm) and expressed in absorbance units (A.U.). At the end of the culture, 1 mM of PMSF (phenylmethylsulphonyl fluoride) as a protease inhibitor, 100 µM of EDTA and 0.05% sodium azide were added to the supernatant. The supernatant was concentrated to a volume of 10 mL using an Amicon^®^ Stirred Cell (Merck™) system and ultrafiltration discs with a nominal molecular weight of 3 kDa (Merck™). It was then filtered again using an Amicon^®^ Ultra-15 Centrifugal Filter Unit (Merck™). The proteins from the cell extract were recovered by sonicating the centrifuged bacterial pellet with 15 pulses of 100 mV (1 min on/1 min off) at 4 °C in an Ultrasonic Processor (Cole-Parmer^®^). To prevent proteolysis, 1 mM of PMSF was added during the sonication. Finally, the concentrated supernatants and cellular extracts of the three strains were recovered and stored at −70 °C until further use.

### 2.6. SDS-PAGE, Immunoblot and Lectin Blot Analysis

The proteins from the cell extracts and supernatants of each strain were quantified using the Bradford Assay (Bio-Rad, Hercules, CA, USA) and 20 µg of each sample was resolved with SDS-PAGE 12% at 100 V for 2 h. For the immunoblot detection, the proteins were transferred to PVDF Amersham™ Hybond^®^ P Western blotting membranes (Amersham Biosciences, Piscataway, NJ, USA) in a Trans-Blot^®^ SD Semi-Dry Transfer Cell (Bio-Rad, Richmond, CA, USA) at 20 V for 30 min. Subsequently, the PstS-1 and Ag85B antigens were immunodetected with TB71 and α-Ag85 monoclonal antibodies (BEI Resources, TBVTRM Contract, Colorado State University, USA), respectively. Briefly, the transferred membrane was blocked with blocking solution (5.0% skimmed milk in TBS-Tween-20, 136 mM NaOH, 2 mM KCl, 25 mM Tris and 0.05% Tween-20) at room temperature (RT) and stirred for 2 h. The membrane was then incubated three times, for 10 min each in TBS-Tween and incubated overnight with the diluted primary antibodies: 1:100 for TB71 and 1:1000 for α-Ag85 in blocking solution at 4 °C with shaking. The membranes were then washed three times with TBS-Tween 20 and incubated with secondary goat anti-mouse IgG (Jackson ImmunoResearch Inc., West Grove, PA, USA) for the PstS-1 membranes and anti-rabbit IgG (ZYMED, Carlsbad CA, USA) for the Ag85B membranes in blocking solution at RT for 2 h with stirring. The membranes were then washed three times for 10 min with TBS-Tween 20. Finally, the membranes were revealed with a substrate diluted with luminol and detected using a chemiluminescent substrate (DuoLuX; Vector Laboratories, Burlingame, CA, USA).

For the lectin blot analysis, 1 µg of the sample proteins was resolved by 12% PAGE, silver-stained and transferred to a PVDF membrane as previously described. The membranes were then analyzed for glycoprotein identification using a DIG Glycan Differentiation Kit™ (Roche, Mannheim, Germany) according to the manufacturer’s instructions. Additionally, glycosylation prediction profiles were performed to evaluate the in silico potential glycosylated residues using predefined algorithms from the bioinformatic software NetOGlyc-4.0 [61], Glycosylation Predictor GPP [62] and GLYCOPP V1.0 [63]. The SDS-PAGE immunoblots and lectin blots were visualized using a GS-800™ Calibrated Imaging Densitometer (Bio-Rad). The image processing and further analysis was performed using the Quantity One™ Image Analysis Software v.4.6 (Bio-Rad, Hercules, CA, USA) and Image J 1.53k.

### 2.7. Protein Labelling and Two-Dimensional Fluorescence Differential Gel Electrophoresis

The extracted proteins from the recombinant *M. microti* strains were fluorescently labeled and analyzed by two-dimensional fluorescence differential gel electrophoresis (2-D DIGE) in duplicate. A CyDye™ DIGE Fluor Labelling Kit (Amersham Biosciences, Buckinghamshire, UK) was used to differentially label the cell extracts of *M. microti*-PMV261, *M. microti*-PstS-1 and *M. microti*-Ag85B with the following dyes: Cy3 (orange), Cy5 (red) and Cy2 (green), respectively. Briefly, the samples were incubated in 400 pmol/µL of dimethylformamide at 4 °C in the dark for 30 min. The reaction was stopped by adding 10 mM of lysine, followed by incubation at 4 °C in the dark for 10 min. The proteins were dissolved in a buffer solution containing 9 M of urea, 4% CHAPS (3-[(3-cholamidopropyl) dimethylammonio]-1-propane sulfonate), 100 mM of dithiothreitol, 0.1% Bio-Lyte^®^ 3/10 Ampholyte (Bio-Rad) and 0.001% bromophenol blue. They were then placed on immobilized pH gradient ReadyStrip™ IPG Strips of pH 4–7, 11 cm (Bio-Rad) for 16 h. The isoelectric focusing was performed under the following conditions: first, a current of 500 V, 1 mA, 5 W, 1 V/h; second, 2500 V, 1 mA, 5 W, 1 V/h; and lastly, 2500 V, 1 mA, 5 W, 49,500 V/h were applied with a MultiPhor™ II Electrophoresis Unit (Pharmacia Biotech, Uppsala, Sweden) within 8 h. Then, the strips were placed in 70 mM of dithiothreitol and a buffer solution composed of 6M of urea, 30% *v/v* glycerol, 50 mM of Tris, 2% SDS and 0.002% bromophenol blue. It was then immediately placed in 193 mM of iodoacetamide. The proteins were separated by electrophoresis in a 12% SDS-PAGE using a SE600 Standard Dual Cooled Vertical Electrophoresis Unit (Hoefer SE600™ unit; Pharmacia Biotech, Uppsala, Sweden) and a buffer solution consisting of 192 mM of glycine, 25 mM pf Tris, and 0.1% SDS at 50 V for 30 min, 100 V for 2 h, and 200 V for 3 h. Finally, the images of the gels were captured using a Typhoon™ 9400 scanner imager (Amersham Biosciences, Piscataway, NJ, USA) and analyzed using the Quantity One™ Image Analysis Software (Bio-Rad, Hercules, CA, USA).

### 2.8. Tandem Mass Spectrometry Analysis

The tandem mass spectrometry (MS/MS) analysis of the selected proteins was performed in an AB Sciex 3200 QTRAP LC/MS System (Applied Biosystems) hybrid tandem mass spectrometer equipped with a Sciex Nanospray II (Applied Biosystems) electrospray ion source and a Sciex Micro Ion Spray Source (Applied Biosystems, Foster City, CA, USA). The proteins were identified based on their MS/MS spectral data sets using the MASCOT v1.6b9 (Matrix Science, London, UK) and the NCBI database previously described [64].

### 2.9. Statistical Analysis

The experiments are presented as mean with standard deviation of at least three independent replicates. The statistical analyses were performed using a one-way analysis of variance (ANOVA) with Tukey’s correction and using the GraphPad Prism software V. 9.0. The differences were considered statistically significant at *p* ≤ 0.05.

## 3. Results

### 3.1. Rv0934 (PstS-1) and Rv1886c (Ag85B) Gene Identification

The primers were designed based on the Rv0934 and Rv1886c genes of *M. tuberculosis*, as described previously, and were used to amplify and compare the same genes of M. microti using PCR. In Figure 1C,D, the PCR amplicons were detected for both strains that corresponded to the appropriate molecular weight (MW) of 1125 bp for Rv0934 and 978 bp for Rv1886c in *M. tuberculosis* (Figure 1C and Figure 1D, lane 1) and *M. microti* (Figure 1C and Figure 1D, lane 2). The sequencing and analysis of the Rv0934 gene from *M. microti* revealed a 99.9% identity to the sequence reported for *M. tuberculosis*, with a base change from T-to-C at nucleotide 1 055 bp, resulting in an Ala-to-Val mutation in the protein sequence (Appendix A). For the Rv1886c gene, we found a 100% identity between the two strains (Appendix A). These results confirmed that the Rv0934 and Rv1886c genes of M. microti are homologous to the previously described genes of *M. tuberculosis*.

### 3.2. Recombinant Strains M. microti-PstS-1 and M. microti-Ag85B Construction

After the successful amplification and sequence analysis of Rv0934 and Rv1886 for both strains, we decided to construct vectors containing the corresponding genes of *M. tuberculosis* in order to overexpress these genes in *M. microti*. The vectors obtained were named pMV261-PstS-1 and pMV261-Ag85B (Figure 1A,B, respectively). After the cloning and propagation of the respective vectors in *E. coli*, the genes were again amplified to confirm their size (Figure 1E,F) and then sequenced (data not shown). Subsequently, the plasmids pMV261-PstS-1 and pMV261-Ag85B were independently cloned into *M. microti*. The resulting recombinant strains were designated *M. microti*-PstS-1 and *M. microti*-Ag85B. Both strains were analyzed for their corresponding gene plasmid transformation. To avoid the genomic amplification of the native genes, a PCR from the isolated plasmid was performed on both *M. microti* recombinant strains (Figure 1G,H). The resulting amplicons from the recombinant *M. microti* strains were verified using sequencing (data not shown). With these results, we confirmed the correct cloning and transformation of the *M. microti* strains.

### 3.3. Kinetic Growth of Recombinant Strains M. microti-PstS-1 and M. microti-Ag85B

After the selection of the positive clones for *M. microti*-PstS-1 and *M. microti*-Ag85B, the strains were cultured in triplicate for 196 h in shake flasks to determine if there were any growth changes due to the genetic loading and to recover any biomass and recombinant protein produced. An *M. microti*-PMV261 strain with the empty plasmid was used as a control in all cases. In Figure 2, the biomass accumulation, measured by optical density (O.D. 600 nm) and expressed in absorbance units (A.U.), is plotted against time (h). The samples were measured every 48 h and the biomass accumulation was plotted on a linear scale (Figure 2A). The biomass obtained after 196 h of culture was 0.71 ± 0.04, 0.76 ± 0.08 and 0.78 ± 0.05 for *M. microti*-PstS-1, *M. microti*-Ag85B and *M. microti* wild type, respectively, with no significant differences compared to the wild-type strain. To compare the specific growth rate (µ), a log-biomass diagram was constructed (Figure 2B) and *µ* was calculated for each strain. We obtained µ values of 0.031 ± 0.002, 0.030 ± 0.005 and 0.035 ± 0.003 for *M. microti*-PstS-1, *M. microti*-Ag85B and *M. microti* wild type, respectively, with no significant differences between the strains. Based on the results, we could conclude that no changes related to the growth of the recombinant *M. microti* strains occurred during the genetic transformation with the constructed vectors.

### 3.4. Production of PstS-1 and Ag85B and Analysis by Western Blotting

An analysis of the PstS-1 and Ag85B proteins in the cell extract and supernatant of the *M. microti* strains was determined through immunodetection with Western blotting using the monoclonal antibodies (mAbs) α-TB71 (PstS-1) and α-Ag85B (Ag85B) after 196 h of culture growth (Figure 3). The results showed that the expression of PstS-1 and Ag85B proteins in the wild-type *M. microti* strain (Figure 3A and Figure 3C, lanes 1 and 4) was at a baseline level in both cellular extracts and supernatants. Surprisingly, the recombinant antigens PstS-1 were detected with MW differences between the cellular extracts (Figure 3A, lanes 1,2 and 3) with a MW of about 32 kDa compared to the secreted ones (Figure 3A, lanes 4, 5 and 7) with a MW of approximately 38 kDa. Moreover, the *M. microti*-PstS-1 strain overexpressed PstS-1 in the cell extract (Figure 3A, lane 3), with a 5.2-fold increase according to the densitometric analysis (Figure 3B), while the release of PstS-1 into the supernatant appeared to be inhibited (Figure 3A, lane 6). In contrast, the *M. microti*-Ag85B strain overexpressed Ag85B in both the cell extract and the supernatant (Figure 3C, lanes 3 and 6). Compared to the *M. microti* wild-type and a control consisting of an *M. microti* with an empty vector (ev), we found a 2.6- and 3.2-fold increase in the Ag85B antigen in the cell extract and supernatant, respectively (Figure 3D). These results show that the overexpression of the PstS-1 (Rv0934) and Ag85B (Rv1886c) antigens of *M. tuberculosis* could be successfully carried out in *M. microti* strains.

### 3.5. Glycosylation of Proteins Produced by M. microti Recombinant Strains

To determine whether *M. microti*-PstS-1, *M. microti*-Ag85B and the *M. microti* wild type can still produce glycosylated proteins, we performed a lectin blot analysis using a DIG Glycan Differentiation Kit™ (Merck™). As suggested by our results, the recombinant *M. microti* strains produced glycosylated proteins (Appendix A). These proteins are likely mannosylated, as we observed signal recognition with GNA lectin. On the other hand, to our surprise, we detected faint bands with an approximate MW of 50 kDa in the three strains when the lectin blot was performed with MAA lectin, which recognizes sialic acids (Appendix A). To our knowledge, this is the first time that sialic acid has been detected in proteins produced by *M. microti* strains. However, a more in-depth analysis of the glycoproteome of *M. microti*, such as MS/MS and the complementation with other techniques, is required to confirm our results. To better understand the glycosylation profile of the PstS-1 and Ag85B antigens, we performed an in silico analysis to predict the potential glycosylation sites on both the recombinant antigens, using three different algorithms to compare the results. In Table 1, we summarize our results of the in silico analysis for the *O*-glycosylation of the PstS-1 and Ag85B proteins.

As our results show, we were able to predict the glycosylation sites of PstS-1 and Ag85B and compare them with the alanine and proline rich antigen APA. PstS-1 showed a positive glycosylation profile after all three analyses. However, differences were observed with respect to the glycosylated residues, as can be seen in Table 1. On the other hand, Ag85B showed positive glycosylation sites only with the Glycosylation Predictor GPP algorithm, with 17 hits. When analyzing the antigen APA and its mannosylation profile with the three algorithms, we also found differences in the number and location of the residue. The mannosylation profile of APA was determined by Dobos et. al. (1996) [65] and later by Smith et. al. (2013) [29], who together reported seven mannosylated residues (Thr 10, 18, 27, 274, 276, 277 and 279). In our results, we found that the most accurate analysis was performed by the GLYCOPP V 1.0 algorithm with six potentially glycosylated residues (Ser 17, Thr 18, 27, 274, 276 and 277). So far, no glycosylation has been demonstrated for Ag85B by other authors, and this was confirmed by our results with GLYCOPP V 1.0. Although PstS-1 has previously been described as a glycolipoprotein, its mannosylation profile has not previously been determined. Our results with GLYCOPP V 1.0 indicate that Thr 86, 160, 331 and Ser 88 are likely to be mannosylated in vivo. Based on our analysis using the antigen APA as a reference for the glycosylation profile, we suggest that prediction of glycosylation with the GLYCOPP V1.0 algorithm could be used to assess a glycosylation profile similar to that occurring in the glycoproteins produced by mycobacteria.

### 3.6. Differential Proteome of M. microti Recombinant Strains

Based on the differences in the PstS-1 production and secretion (Figure 3A, lanes 3 and 6), we decided to perform a proteomic analysis to investigate what other changes might occur in native proteins in the *M. microti*-PstS-1 and *M. microti*-Ag85B recombinant strains and compare them with the *M. microti* wild type. The two-dimensional fluorescence differential gel electrophoresis was performed with labelled proteins from the cell extracts of the three strains, as mentioned in materials and methods. On the fluorescence images obtained for each strain, we observed different protein spots that were differentially expressed compared to each other: *M. microti* vs. *M. microti*-PstS-1, *M. microti* vs. *M. microti*-Ag85B and *M. microti*-PstS-1 vs. *M. microti*-Ag85B (Appendix A). The image analysis allowed us to perform a silver stained 2D SDS-PAGE (Figure 4) to identify the spots that were differentially expressed and to select the spots with the highest intensities for further sequencing using MS/MS. Our analysis results showed that 46 proteins were differentially expressed in the three examined *M. microti* strains and based on intensity, we selected 18 proteins for further analysis. These results suggest that changes in the genetic load of *M. microti*, such as the insertion of foreign plasmids and the subsequent overexpression of proteins, cause changes in the proteome profile of *M. microti*-PstS-1 and *M. microti*-Ag85B in the cell extract. However, further high throughput proteome and transcriptome analyses are still required to substantiate our results.

### 3.7. MS/MS Identification of Differentially Expressed Proteins in M. microti Strains

The selected protein spots based on the intensity values from Figure 4 were analyzed and sequenced using MS/MS. The spectra data sets were processed using MASCOT v1.6b9 (Matrix Science) and BLAST from the NCBI database. The identified proteins are summarized in Table 2, together with their unique accession numbers from the UniProt database [66] and a brief description of each identified protein. Our results show that spots 2, 3, 4, 5, 6, 7, 10, 11, 13, 15, 16, 17, 18, 19 and 20 correspond to either enzymes or enzyme subunits, while spot 22 is involved in protein folding and spots 12 and 21 have an unknown function. Post-translational modifications (PTM) were present in spots 10, 12, 15, 16, 17, 20, 21 and 22 according to mass spectra data (not shown). These PTM’s correspond to mannose glycans, as previously noted in our lectin blot analysis. As for the differential expression between the recombinant *M. microti* strains, we found that spots 2, 3, 4, 5, 6, 7, 11, 12, 13, 18 and 19 showed a differential expression between the three strains either by presence (+) or absence (-), as indicated in Table 2. Based on the differential expression of the three *M. microti* strains, a Venn diagram was constructed (Figure 5). Considering the different proteins present only in *M. microti*-PstS-1 and *M. microti*-Ag85B (MoxR, RpoA, Eno, SodA, DapA and MAP_3475c) and related to energy production and growth, we can expect to observe the overexpression of these types of proteins to cope with the metabolic burden of a recombinant protein.

## 4. Discussion

Technologies are currently being used to develop new vaccines and/or treatments to prevent TB, which is a global health priority. One of these is the use of proteomics and genetic engineering to discover and produce antigens with a known ability to elicit an immune response [67]. Our work focused on the genetic engineering of mycobacterial strains belonging to the Mtb complex to produce antigens of interest. Such antigens and their epitopes, either alone or in combination, may help to develop new ways to increase the efficacy of current TB vaccines, for example, the currently available *M. bovis* BCG live-attenuated vaccine [68,69,70,71,72]. Therefore, we generated and analyzed two recombinant strains of *M. microti* overexpressing the *M. tuberculosis* PstS-1 and Ag85B antigens. Both antigens are known to stimulate the immune system and act as potent immunomodulators in the production of cytokines and antibodies when used as multimeric vaccine components [37,68,70,71]. The complete genome of *M. microti* has not been released up to date. However, studies show that the RD1 region is absent in several *M. microti* strains [73]. In our work, a comparative genetic analysis of Rv0934 (PstS-1) and Rv1886c (Ag85B) revealed that these genes are present in our wild-type *M. microti* (ATCC 11152) strain with a homology of 99.9% and 100%, respectively. This confirms that these selected antigens from *M. microti* are very similar in sequence to those previously described in *M. tuberculosis* and may have a similar immunomodulatory function.

PstS-1 and Ag85B were overexpressed and detected in both *M. microti* strains (Figure 3). For the PstS-1 antigen, we observed differences in the MW size depending on whether the protein originated from a cell extract or from the secreted form. First, an analysis of the cellular extract of PstS-1 revealed a band showing a 5.2-fold increase in production of the antigen with a MW of ~32 kDa (Figure 3, lane 3). However, we did not observe a secretion of PstS-1 in this strain (Figure 3, lane 6). Instead, we detected a band size of 38 kDa in *M. microti* (wt) and *M. microti* (ev) (Figure 3, lanes 4 and 5). We hypothesize that this difference in the molecular mass of this protein is due to the detection of proteins prior to the post-translational modifications, as PstS-1 typically undergoes post-translational modifications, including glycosylation and lipoylation, which likely contribute to its molecular mass [36]. In addition, the glycosylation of *M. tuberculosis* proteins may also represent a mechanism for regulating protein export [74]. It is possible that the recombination influenced the transport of this protein outside the bacterial cell. However, this is not a problem, as the intracellular antigen could still be available through the processing performed using professional antigen-presenting cells, where the phagosome maturation leads to the lysis of the microorganism and the antigens are subsequently presented by the molecules of the major histocompatibility complex (MHC) [75,76,77]. On the other hand, the expression analysis of Ag85B showed that this antigen was successfully expressed in both the cellular and secreted forms. Western blotting showed the presence of the three bands (Figure 3C lanes 1–6) in both the cell extract and the supernatant. This is probably due to the fact that the antibody used in this study can detect the Ag85 complex (A, B and C) in addition to Ag85B (Figure 3C, lanes 1–6 compared to control lane 7).

These results led us to investigate whether the recombinant strains had different proteomes and/or glycosylation profiles. We found that each strain of *M. microti*-PstS-1 and *M. microti*-Ag85B had a different proteomic profile. We identified six proteins (SodA, hypothetical protein MAP3475c, DapA, MoxR, RpoA and Eno) involved in the energy production process, possibly due to an increased heterologous protein production, that are not expressed in the *M. microti* wild type. Similar observations were previously made by Cheng et al. who reported that the overexpression of proteins in recombinant *Escherichia coli* strains leads to a change in the protein profiles, resulting in a differential expression of different cellular proteins [78,79].

There are few studies describing the presence of glycosylation in the proteins of the genus *Mycobacterium*, and most existing studies have focused on mannosylation [54,80,81]. Examining the glycosylation profiles in the wild-type and the recombinant *M. microti* allowed us to identify two potential glycosylation types in these strains: mannosylation and sialylation, as shown in Appendix A. Mannose glycosylation is widely reported in the genus *Mycobacterium* [49,50,54,82]. In the present study, we identified and sequenced eight proteins that were potentially glycosylated with terminal mannose, including MAP_4308c, hypothetical protein p40, MAP_3192, FabG, GlyA, Eno, MAV_0024 and GroEL. The mannosylation has various functions in mycobacteria. For example, it is part of the cell wall and is primarily associated with virulent species and enables protein recognition by mannose and the Toll-like receptor (TLR) 2 expressed by macrophages and other cell types [83]. In addition, studies of proteins that were glycosylated with mannose, such as PstS-1 from *M. tuberculosis* H37Rv, have shown that these proteins promote phagocytosis [35]. Pan et al. developed ZXL1 (double-stranded DNA), a therapeutic target that prevents mannose from binding to host cells by inhibiting the immunosuppression of CD11c+ dendritic cells and enhancing the activity of antigen-presenting cells for the activation of naïve CD4+ Th1 cells [84]. For PstS-1 and Ag85B, we predicted the glycosylation profiles using three algorithms that allowed us to compare the different results in terms of the number and position of residues involved in mannosylation. Our results show that PstS-1 could be produced in its mannosylated form. In contrast, no glycosylation was predicted for Ag85B (Table 1) using the GLYCOPP V 1.0 algorithm.

We have identified proteins that may be glycosylated with sialic acid (Appendix A). Currently, there is little evidence of proteins glycosylated with sialic acid in the genus Mycobacterium. Therefore, we searched the genome of *M. microti* for enzymes associated with sialic acid metabolism that have been identified in other bacteria, including *Neisseria meningitides*, *E. coli* K12 and *Haemophilus influenzae* [85,86,87,88]. We found that *M. microti* contains gene encoding proteins that are known to be involved in the de novo synthesis of sialic acid (UDP–*N*–acetylglucosamine–pyrophosphorylase/glucosamine–1–phosphate-*N*-acetyltransferase, l-serine dehydratase and the hypothetical protein B8A25_03135) in the case of *M. tuberculosis sp.* variant *microti* (data not shown). Overall, these results remain to be confirmed by mass spectrometry assays. The identification of the differences in the proteomic and glycosylation profiles between the wild type and recombinant strains of *M. microti* may lead to alterations in the immune system recognition and evasion [89,90]. For example, the mannose glycosylation of the protein APA has been shown to play a key role in its antigenicity, while the deglycosylation of the same protein has been demonstrated to modify the immune response elicited in guinea pigs [53]. Taken together, these results demonstrate the need to thoroughly investigate the proteomic profiles of the different mycobacterial strains being considered for the development of new TB vaccines, as these events may alter the immune response and the effectiveness of these potential vaccines.

## 5. Conclusions

This study has shown that the overexpression of PstS-1 and Ag85B in the recombinant strains of *M. microti* alters their proteome, suggesting that the impact of these changes on the immune responses elicited by these and other newly generated strains, needs to be re-evaluated, especially when considering their potential use in the development of TB vaccines. In addition, we observed a differentially mannosylated profile in the analyzed proteome and we obtained evidence for the possible sialylation of the proteins expressed by *M. microti*, although this remains to be confirmed. These glycosylations play a key role in the interactions between bacteria and host cells. Therefore, studying the role of these PTMs in this genus is of the utmost importance to understand how these modifications affect their immunogenicity.

## Figures and Tables

**Figure 1 biomolecules-12-01836-f001:**
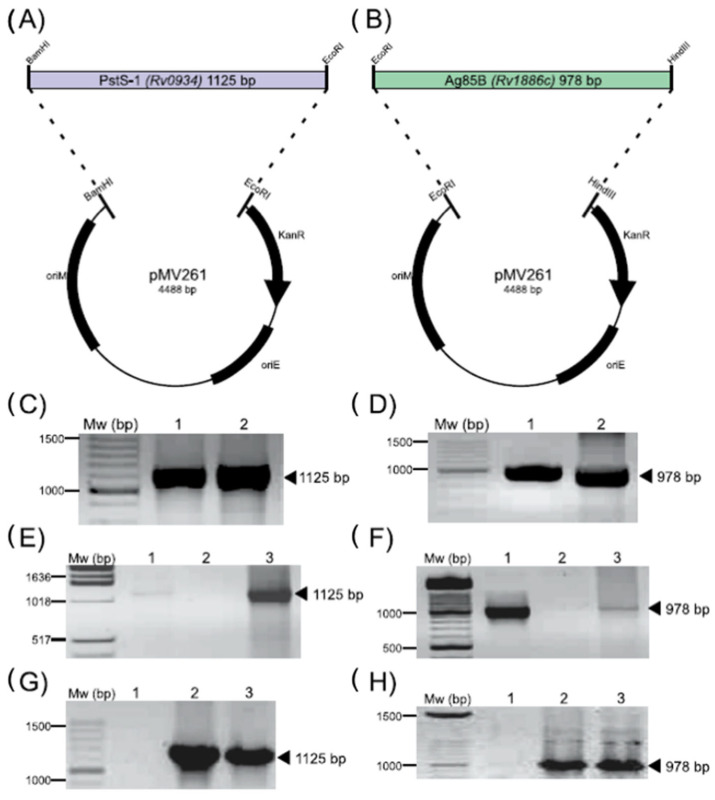
Plasmid design and construction of (**A**) pMV261-Rv0934 for PstS-1 and (**B**) pMV261-Rv1886c for Ag85B. (**C**,**D**) PCR amplification of Rv0934 and Rv1886c, respectively, from *M. tuberculosis* (lane 1) and *M. microti* (lane 2). (**E**,**F**) PCR amplification of Rv0934 and Rv1886c from plasmid isolated from *E. coli* colonies after transformation with pMV261-Rv0934 and pMV261-Rv1886c, respectively (lane 3); controls (lanes 1 and 2, respectively). (**G**,**H**) PCR amplification of Rv0934 and Rv1886c from plasmid isolated from *M. microti* colonies transformed with pMV261-Rv0934 and pMV261-Rv1886c, respectively (lane 3); controls (lanes 1 and 2). Black arrows indicate the expected amplicon size.

**Figure 2 biomolecules-12-01836-f002:**
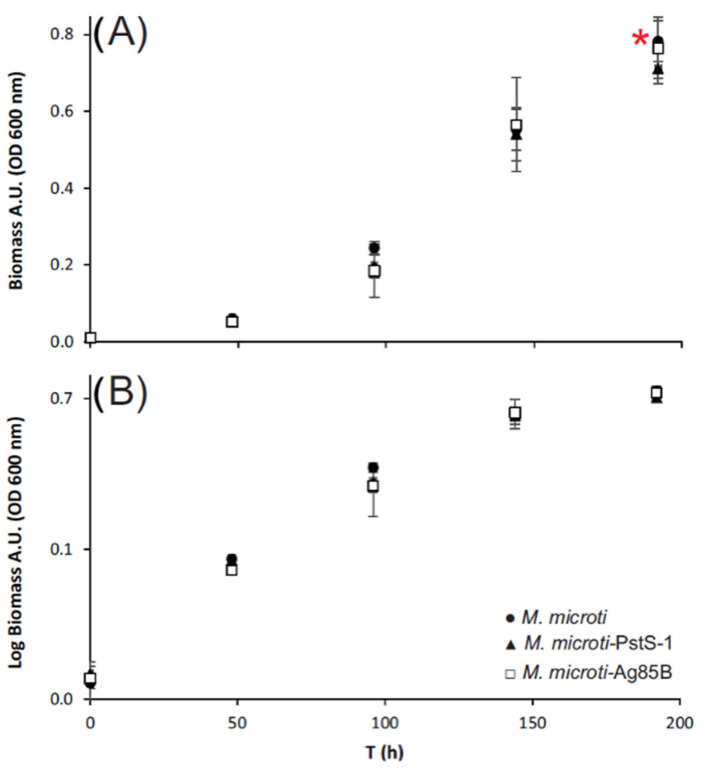
Kinetic growth of *M. microti* strains in Sauton media. The biomass accumulation in absorbance units (A.U.) measured by optical density (O.D.) during 196 h: (**A**) Lineal scale growth and (**B**) Log scale growth of the strains *M. microti* (closed circles), *M. microti*-PstS-1 (closed triangles) and *M. microti*-Ag85B (open squares). The red asterisk indicates the collection time for protein analysis.

**Figure 3 biomolecules-12-01836-f003:**
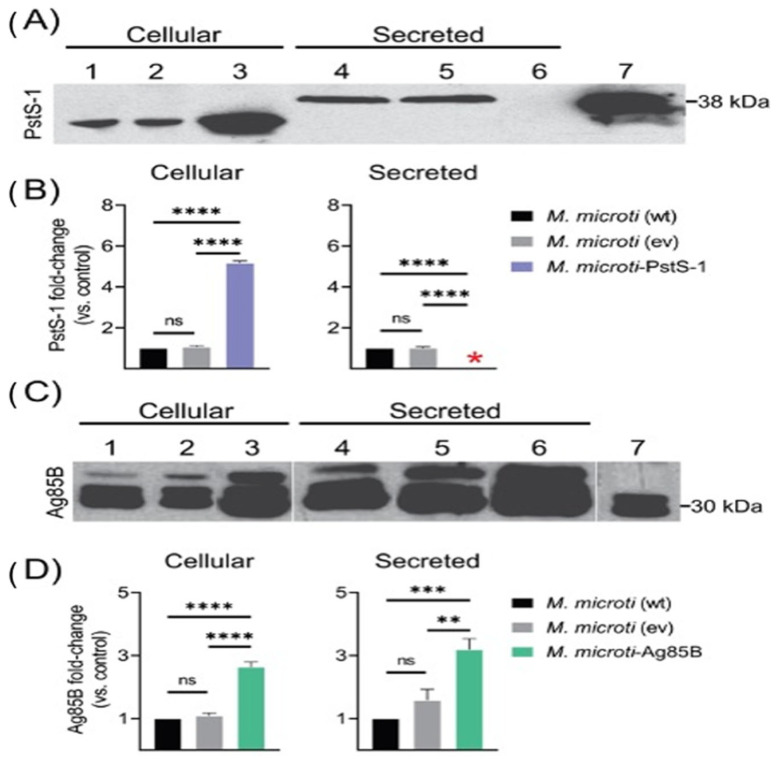
Protein analysis of PstS-1 and Ag85B, either cellular or secreted, produced by *M. microti* strains. (**A**,**C**) detection by Western blot of antigens PstS-1 and Ag85B, respectively. (**B**,**D**) fold expression analysis of PstS-1 and Ag85B, respectively. (**A**,**C**) lanes 1 and 4: *M. microti* wild type (wt), lanes 2 and 5: *M. microti* with empty vector (ev), lane 7 purified protein control. (**A**) lanes 3 and 5 *M. microti*-PstS-1; (**B**) lanes 3 and 5 *M. microti*-Ag85B. The fold change analysis performed by densitometric analysis by Image J. Red asterisk indicates the absence of protein. The differences were considered statistically significant with a *p* ≤ 0.05. **, ***, ****, ns = not significant. Using a one-way analysis of variance (ANOVA) with Tukey’s correction.

**Figure 4 biomolecules-12-01836-f004:**
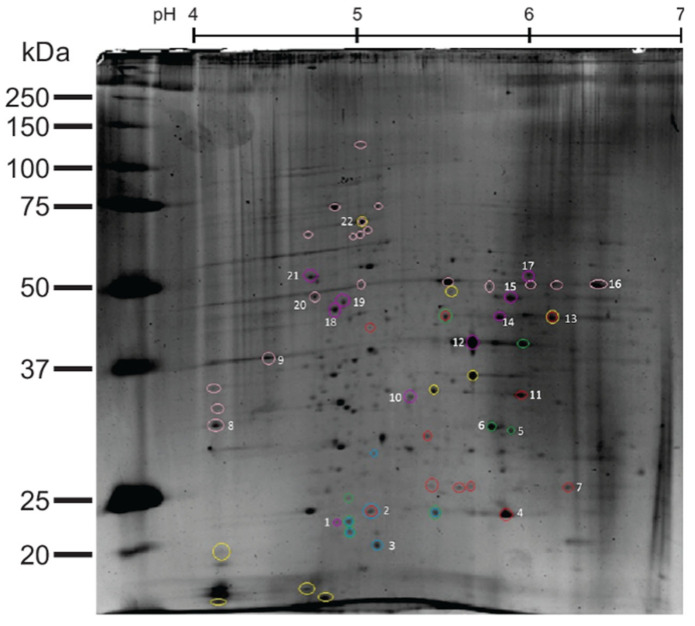
2D SDS-PAGE cell extract proteome analysis of *M. microti* strains. Circled proteins were selected based on the fluorometric analysis according to the differential expression: *M. microti*-PMV261 (red circles), *M. microti*-PstS-1 (blue circles) and *M. microti*-Ag85 (green circles). Numbered proteins were sequenced using MS/MS (Table 2). Unnumbered pink and yellow circled spots represent glycosylated proteins.

**Figure 5 biomolecules-12-01836-f005:**
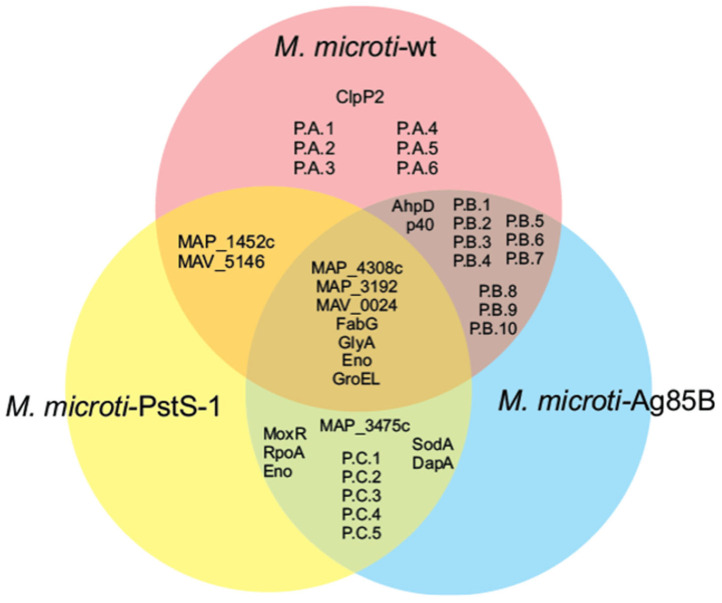
Venn diagram of the differential expression proteome of the cellular extract from *M. microti* strains. P.A, P.B and P.C: protein spots from Appendix A–C, respectively.

**Table 1 biomolecules-12-01836-t001:** Potential glycosylated residues in recombinant antigens PstS-1 and Ag85B.

Antigen	UniProt Acc. No.	NetOGlyc-4.0 ^1^	Glycosylation Predictor GPP ^2^	GLYCOPP V 1.0 ^3^	Experimental Determination
PstS-1 (Rv0934)	B2MVV3	**S30**, **S32**, T35, T41, **T44**, **T45**, **S48**, **S49**, **S290**	T7, T13, S26, S30, S32, **T44**, **T45**, **S48**, **S49**, T52, T56, T59, T80, T99, S110, S128, S140, **T160**, S186, T196, T216, T236, **S290**, S299, **T331**, S365, T371, S373, S374	T86, S88, **T160**, **T331**	ND
Mod. AA		9	29	4	
Ag85B (Rv1886c)	P9WQP1	None predicted	T2, S5, S42, S57, N71, S116, S117, S120, S150, T158, S160, S166, S171, S202, T234, S320, S321	None predicted	ND
Mod. AA		0	17	0	
APA (Rv1860)	P9WIR7	**T10**, **T11**, **S14, S17**, **T18, T27**, T39, **T108**, S243, **T274**, **T276**, T277, T279, **T283**	**T10**, **T11**, **S14**, **S17, T18**, **T27**, S101, S105, T107, **T108**, T142, S144, S152, T164, S181, S183, S190, S193, T201, S243, **T274**, **T276**, **T283**	**S17**, **T18**, **T27**, **T274**, **T276**, **T277**	T10, T18, T27, T277 (Dobos et al., 1996 [65]) T274, T276, T277, T279 (Smith et al., 2014 [29])
Mod. AA		14	23	6	4

Acc. No.: Accession number, Mod. AA: Modified amino acids, ND: Not determined. APA antigen was used for comparison purposes and analysed in silico without the signal peptide composed of residues 1–39. Residues in bold are shared by two or more prediction tools. ^1^ Based on a neural network predictions of mucin type GalNAc O-glycosylation sites in mammalian proteins where a value higher than 0.5 was considered a positive hit [61]. ^2^ Prediction based on random forests algorithms [63]. ^3^ Prediction based on Composition Profile of Patterns (CPP) where a value higher than 0.9 was considered a positive hit [62].

**Table 2 biomolecules-12-01836-t002:** Proteins identified by MS/MS from *M. microti* strains.

Spot	Protein	UniProt	Function	MW (kDa)	Global Score *	*M. microti* strains			PTM
						wt	PstS-1	Ag85B	
**2**	ATP-dependent Clp protease proteolytic subunit 2 (ClpP2) ^1^	Q73XM8	Cleaves peptides in various proteins in a process that requires ATP hydrolysis. Has a chymotrypsin-like activity. Plays a major role in the degradation of misfolded proteins.	21.665	297	+	-	-	ND
**3**	Alkyl hydroperoxide reductase (AhpD) ^1^	Q73ZL4	Antioxidant protein with alkyl hydroperoxidase activity. Required for the reduction of the AhpC active site cysteine residues and for the regeneration of the AhpC enzyme activity.	18.842	151	+	-	+	ND
**4**	Superoxide dismutase (SodA) ^2^	B1A036	Destroys superoxide anion radicals which are normally produced within the cells, and which are toxic to biological systems.	22.476	125	-	+	+	ND
**5**	AB hydrolase-1 domain-containing protein (MAP_1452) ^1^	Q73ZZ9	Hydrolase activity.	32.404	303	+	+	-	ND
**6**	Peroxisomal multifunctional enzyme type 2 (MAV_5146) ^3^	A0A0H2ZVK0	3-hydroxyacyl-CoA dehydrogenase activity. 3alpha, 7alpha, 12alpha-trihydroxy-5beta-cholest-24-enoyl-CoA hydratase activity.	29.906	945	+	+	-	ND
**7**	Nitroreductase domain-containing protein (MAP_3475c) ^1^	Q73U93	Oxidoreductase activity: nitroreductase	24.012	186	-	+	+	ND
**10**	Fructose-bisphosphate aldolase class 1 (MAP_4308c) ^3^	Q73RX0	Fructose-bisphosphate aldolase activity.	33.645	891	+	+	+	Yes
**11**	4-hydroxy-tetradihydrodipicolinate synthase (DapA) ^1^	Q73VZ7	Catalyzes the condensation of (S)-aspartate-beta-semialdehyde [(S)-ASA] and pyruvate to 4-hydroxy-tetrahydrodipicolinate (HTPA).	30.973	350	-	+	+	ND
**12**	p40 protein ^4^	Q9AIQ0	ND	36.232	1109	+	-	+	Yes
**13**	ATPase (MoxR) ^1^	Q740Y7	ATP binding and hydrolysis activity	40.730	308	-	+	+	ND
**15**	AB hydrolase-1 domain-containing protein (MAP_3192) ^1^	Q73V24	Hydrolase activity	43.256	674	+	+	+	Yes
**16**	3-ketoacyl-(acyl-carrier-protein) reductase (FabG) ^5^	I2AKG1	3-oxoacyl-[acyl-carrier-protein] reductase (NADPH) activity.	47.139	1008	+	+	+	Yes
**17**	Serine hydroxymethyltransferase (GlyA) ^1^	Q73WG1	Catalyzes the reversible interconversion of serine and glycine with tetrahydrofolate (THF) serving as the one-carbon carrier.	44.955	46	+	+	+	Yes
**18**	DNA-directed RNA polymerase subunit alpha (RpoA) ^1^	Q73S43	DNA-dependent RNA polymerase catalyzes the transcription of DNA into RNA using the four ribonucleoside triphosphates as substrates.	37.704	1261	-	+	+	ND
**19**	Enolase (Eno) ^1^	Q741U7	Catalyzes the reversible conversion of 2-phosphoglycerate into phosphoenolpyruvate. It is essential for the degradation of carbohydrates via glycolysis.	44.873	1242	-	+	+	ND
**20**	Enolase (Eno) ^3^	A0QBX4	Catalyzes the reversible conversion of 2-phosphoglycerate into phosphoenolpyruvate. It is essential for the degradation of carbohydrates via glycolysis.	44.873	1096	+	+	+	Yes
**21**	Xyppx repeat family protein (MAV_0024) ^3^	A0A0H3A0B1	ND	58.801	214	+	+	+	Yes
**22**	Chaperonin (GroEL) 2 ^3^	A0QLP6	Together with its co-chaperonin GroES, plays an essential role in assisting protein folding.	56.615	17263	+	+	+	Yes

Detected peptides were matched against reference sequences: ^1^
*Mycobacterium avium* spp. *paratuberculosis* K10, ^2^ *M. avium* spp. *paratuberculosis*, ^3^ *M. avium* 104, ^4^
*M. avium*, ^5^ *Mycobacterium* spp. MOTT36Y. * Scores >25 are considered significant (*p* < 0.05) according to the Mascot Search Results algorithm. MW: molecular weight, kDa: kilodalton, WT: wild type PTM: post-translational modified, ND: not detected/determined.

## Data Availability

Not applicable.

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
