# Peer review of "Proteome Profile Changes Induced by Heterologous Overexpression of Mycobacterium tuberculosis-Derived Antigens PstS-1 (Rv0934) and Ag85B (Rv1886c) in Mycobacterium microti"

_biomolecules, 2022, doi:10.3390/biom12121836_

Round 1
Reviewer 1 Report
There are many errors. For example, "Gene extraction and sequencing ", "The protein expression of both antigens was driven by a constitutive vector", Middlebrook 7H10 Agar with 30 g/mL of kanamycin".
"non-transformed M. microti strain was used as a control in all cases" - comparison of proteomic profiles of strains with antibiotic resistance should be vs strain with an empty plasmid
Author Response
We greatly appreciate all suggestions and observations regarding our manuscript.
In relation to the requested changes in our research article “Proteome profile changes are induced by heterologous overexpression of Mycobacterium tuberculosis-derived antigens PstS-1 (Rv0934) and Ag85B (Rv1886c) in Mycobacterium microti”, we made the following changes
Reviewer 1 comments:
There are many errors. For example, "Gene extraction and sequencing ", "The protein expression of both antigens was driven by a constitutive vector", Middlebrook 7H10 Agar with 30 g/mL of kanamycin".
"non-transformed M. microti strain was used as a control in all cases" - comparison of proteomic profiles of strains with antibiotic resistance should be vs strain with an empty plasmid.
Answer: We appreciate your suggestions then; we have checked the errors in the manuscript and corrected them accordingly with the correct wording. For example, we changed “gene extraction and sequencing” to “DNA extraction and sequencing”.
We made a mistake during the writing of the manuscript, since the strain used was the strain with the empty plasmid. The information was corrected in the text, which now reads as follows: “A M. microti-PMV261 strain with the empty plasmid was used as a control in all cases”
In addition, we verify the units in the reagents we use in this study since the format has been lost in PDF format. Nevertheless, we checked the units and changed them appropriately.
Reviewer 2 Report
In the manuscript “Proteome profile changes are induced by heterologous overexpression of Mycobacterium tuberculosis-derived antigens PstS-1 (Rv0934) and Ag85B (Rv1886c) in Mycobacterium microti” by Viridiana García-Ruiz and colleagues, the authors generated and analyzed two recombinant strains of M. microti overexpressing the M. tuberculosis (Mtb) PstS-1 and Ag85B antigens. The authors observed a differentially mannosylated profile on the proteome and obtained evidence that could indicate the possible sialylation of proteins expressed by M. microti, although not showing evidence to support this claim. Finally, the authors conclude that the role of post-translational modifications in Mycobacteria is of paramount importance to understand how these modifications impact their immunogenicity, which can bring important lessons for vaccine design. The study was well conducted, with the appropriate controls in place, well presented and the discussion covers all the relevant details. The results are clear, the conclusions well supported, and the claims presented here are relevant. I have no major concerns about this manuscript.
Minor concerns and comments:
1) Given that the Mtb antigens studied in this manuscript are also naturally expressed in M. microti, could the authors please elaborate on the advantage of using these particular antigens in a potential M. microti recombinant vaccine?
2) Please describe the promoter that was used in the constructs, to express the Mtb genes.
3) In the figure 3 B and 3 D, both panels, the middle bar should have ns or stars. Please add the relevant symbol.
4) In the legend of figure 3, please add the values of the statistical significance observed.
5) Line 226, the authors used 1 g of DNA for electroporation of competent M. microti cells. Could the authors please verify this and include the correct units?
6) Line 233 and 273, please verify and include the correct units.
7) Please remove lines 592 to 594.
Author Response
We greatly appreciate all suggestions and observations regarding our manuscript.
In relation to the requested changes in our research article “Proteome profile changes are induced by heterologous overexpression of Mycobacterium tuberculosis-derived antigens PstS-1 (Rv0934) and Ag85B (Rv1886c) in Mycobacterium microti”, we made the following changes:
Minor concerns and comments:
- Given that the Mtb antigens studied in this manuscript are also naturally expressed in M. microti, could the authors please elaborate on the advantage of using these particular antigens in a potential M. microti recombinant vaccine?
Answer: These antigens are well described and play an important role in the immune response or as biomarkers. For example, PstS-1 antigen amplifies IFN-γ and induces IL-17/IL-22 responses by unrelated memory CD4+ T cells via dendritic cell activation, PstS1 also is a highly immunogenic and immunostimulatory component of the mycobacterial cell membrane and a good candidate for the diagnosis and vaccination against tuberculosis. On the other hand, Ag85B, is one of the most abundant proteins in culture of Mycobacterium tuberculosis and is highly immunogenic, as shown by the easy detection of specific humoral and cell-mediated immune responses in both latently and actively infected TB patients. In addition, this antigen has also been studied as an adjuvant in some other vaccines against other diseases.
In addition, we used the Mycobacterium tuberculosis antigens since the genome of several strains of Mycobacterium tuberculosishas been published, while the genome of Mycobacterium microti has not yet been published.
- Please describe the promoter that was used in the constructs, to express the Mtb genes.
Answer: The promoter of the plasmid PMV261 is Hsp60 and the information was added in the line 218.
- In the figure 3 B and 3 D, both panels, the middle bar should have ns or stars. Please add the relevant symbol.
Answer: The symbols were added to the Figure 3B and 3D, page 10.
- In the legend of figure 3, please add the values of the statistical significance observed.
Answer: The values of the statistical significance was added in the legend of the figure 3.
- Line 226, the authors used 1 g of DNA for electroporation of competent M. microti cells. Could the authors please verify this and include the correct units?
Answer: The units were reviewed and corrected in all the manuscript.
- Line 233 and 273, please verify and include the correct units.
Answer: The units were reviewed and corrected in all the manuscript.
- Please remove lines 592 to 594.
Answer: We removed the lines 563 to 565 in which they mentioned the instructions for the “Conclusion” section. We did not delete the lines 592 to 594 as in those lines it’s shown the follow information:
Yolanda López-Vidal. All authors read and approved the final manuscript. 592
Conflicts of Interest Statement: The authors do not have any conflicts of interest to declare. 593
References 594

Reviewer 3 Report
This article is covering some aspects of the Proteome profile changes by induction of overexpression of Mycobacterium tuberculosis-derived antigens PstS-1 in Mycobacterium microti.
The specific aims of this article are exclusively directed on differences in the glycosylation profiles of two recombinant strains, which were constructed with overexpression of antigens Ag85B and PstS-1. The authors proved the presence of mannosylated proteins and sialic acid linked proteins.
This will constitute the important goals and novelty of this paper. The article is concluded with a collection of 89 mostly recent references.
Additionally, all 5 figures in the “3.Results” section are very informative and with concise important data comparison summarizing the interplay between those antigens.
The following suggested changes and recommendations should be introduced before the publication of the manuscript.
1. General remarks; All 5 figures in the results section are missing sequential numbering and consequently ‘Legend text “ describing the importance of presentation. This should be corrected
accordingly.
2. Page 2. Line 86. The other proteins such PPE 51 involved in transport should be listed and referenced here and cited in references!! See; Cells 2020, 9, 603; doi:10.3390/cells9030603
3. Page 7. Line 328. The text is referred to figure 1C and 1D. It is very confusing without a real legend for Figure 1. This must be corrected !!!
4. Page 9. Line 374. The same as in 3. This must be corrected!!
5. Page 13. Line 488. (Figure 3) Where is the Figure 3?? And its legend??
6. Page 15. Line 575. The words “especially interactions with the immune system” should be removed, as there is no discussion and references to the immune system in the text.
The manuscript is of good quality and importance and is sequentially written and edited in order to meet the standard for the articles published inBiomolecules. Thus, I certainly recommend it for publication after the correction of these suggested minor changes and recommendations.
Author Response
We greatly appreciate all suggestions and observations regarding our manuscript.
In relation to the requested changes in our research article “Proteome profile changes are induced by heterologous overexpression of Mycobacterium tuberculosis-derived antigens PstS-1 (Rv0934) and Ag85B (Rv1886c) in Mycobacterium microti”, we made the following changes:
Reviewer 3 comments:
- General remarks; All 5 figures in the results section are missing sequential numbering and consequently ‘Legend text “describing the importance of presentation. This should be corrected
Answer: The numbers of the figures were reviewed and corrected accordingly with the text. The “Legend text” appears in a document apart assigned for it.
- Page 2. Line 84. The other proteins such PPE 51 involved in transport should be listed and referenced here and cited in references!! See; Cells2020, 9, 603; doi:10.3390/cells9030603
Answer: We appreciate your suggestion and we included in our manuscript list, Then, in paragraph
- Page 7. Line 328. The text is referred to figure 1C and 1D. It is very confusing without a real legend for Figure 1. This must be corrected!!!
Answer: The “Legend text” and the description of the images is in the document are described in the document apart named “Figure legends”
- Page 9. Line 374. The same as in 3. This must be corrected!!
Answer: The “Legend text” and the description of the images is in the document are described in the document apart named “Figure legends”
- Page 13. Line 488. (Figure 3) Where is the Figure 3?? And its legend??
Answer: The sequence of the figures was corrected in the text and the information for each figure are described in the document apart named “Figure legends”
- Page 15. Line 575. The words “especially interactions with the immune system” should be removed, as there is no discussion and references to the immune system in the text.
Answer: The sentence was removed in the manuscript.
Once again, thank you very much for the time for review the manuscript and their comments.

Round 2
Reviewer 1 Report
The aim of this work was to determine the differences in the glycosylation profiles of two recombinant strains of Mycobacterium microti overexpressing-22 M.tuberculosis Ag85B (Rv1886c) and PstS-1 (Rv0934) antigens. The study of post-translational modifications in various mycobacterial strains is important for the development of effective vaccines for the prevention of tuberculosis.